# Telomere-to-Telomere Gap-Free Genome Assembly and Comparative Analysis of the *Opsariichthys bidens* (Cypriniformes: Xenocyprididae)

**DOI:** 10.3390/biology14111544

**Published:** 2025-11-03

**Authors:** Xinyue Wang, Qi Liu, Denghua Yin, Pan Wang, Min Jiang, Jie Liu, Ning Sun, Yunzhi Yan, Kai Liu

**Affiliations:** 1School of Ecology and Environment, Anhui Normal University, Wuhu 241000, China; xinyuewang@ahnu.edu.cn (X.W.); sunning02260@163.com (N.S.); 2Key Laboratory of Freshwater Fisheries and Germplasm Resources Utilization, Ministry of Agriculture and Rural Affairs, Freshwater Fisheries Research Center, Chinese Academy of Fishery Sciences, Wuxi 214081, China; yindenghua@ffrc.cn (D.Y.); wangmufen1219@163.com (P.W.); jiangmin@ffrc.cn (M.J.); maydayjiel@163.com (J.L.); 3College of Fisheries, Southwest University, Chongqing 400715, China; liuqi_agr@163.com

**Keywords:** *Opsariichthys bidens*, T2T, gap-free, genome assembly, stream environment, adaptation

## Abstract

**Simple Summary:**

*Opsariichthys bidens* is a widely distributed freshwater fish uniquely adapted to the challenging conditions of stream environments, which are typically characterized by high dissolved oxygen and sustained high-flow velocities. To thrive in such habitats, stream fishes often evolve enhanced capabilities in oxygen utilization, energy metabolism, and sustained swimming performance. In this study, we constructed the telomere-to-telomere (T2T) reference genome for *O. bidens*. Genomic analyses identified several expanded gene families and pathways associated with these adaptive traits. This high-quality genome assembly provides a foundational resource for further research into the genetic mechanisms underlying ecological adaptation in stream-dwelling fishes.

**Abstract:**

Stream-dwelling fishes face diverse hydrological pressures, making the broadly distributed *Opsariichthys bidens* an ideal model for analyzing adaptive evolution. To elucidate its adaptation to a high-dissolved-oxygen and high-flow-velocity stream environment, a high-quality genome with comprehensive annotation is essential. In this study, we present the first telomere-to-telomere (T2T) reference genome for *O. bidens*, constructed using PacBio HiFi, Oxford Nanopore Ultra-long, and Hi-C technologies. The assembled genome spans 841.96 Mb, comprising 38 chromosomes, each in a single contig (contig N50 = 22.42 Mb, 2.5-fold higher than the previous version), achieving a gap-free standard with 99.34% BUSCO completeness. Additionally, 38 centromeric sequences, 37 double-telomeric sequences, and 1 single-telomeric sequence were successfully identified, providing essential molecular markers. Phylogenetic analysis revealed a divergence time of 13.5 million years between *O. bidens* and its closely related species *Z. platypus*, with collinearity analysis confirming their high genomic conservation. Gene family analysis revealed 350 expanded families enriched in pathways associated with adaptation to high-dissolved-oxygen environments (e.g., antioxidant defense, oxidative phosphorylation, mitochondrial electron transport chain) and high-flow-velocity environments (e.g., exercise endurance, myocardial contraction, actin binding). Positive selection analysis further identified multiple pathways and key genes involved in mitochondrial optimization, oxygen utilization, and metabolic regulation. The T2T assembly greatly improves assembly continuity and enabling precise identification of centromeres and telomeres for *O. bidens*. These results provide a robust foundation for studying its adaptive evolution to stream environment.

## 1. Introduction

*Opsariichthys bidens* belongs to the order Cypriniformes, family Xenocyprididae, and genus *Opsariichthys* [1,2]. This species exhibits a broad geographic distribution across China, ranging from Hainan Island in the south to Heilongjiang in the north, the Sichuan Basin in the west, and the eastern coastal regions. *O. bidens* is a typical stream-dwelling fish, preferring fast-flowing, oxygen-rich clear streams. Due to its wide ecological adaptability, high trophic level, and early phylogenetic position, *O. bidens* represents a key species for investigating the structure and function of stream ecosystems [3].

The stream environment is characterized by typical hydrological dynamics, including high flow velocity, elevated dissolved oxygen levels, and seasonal fluctuations in water volume [4]. This complex environment presents significant challenges for the ecological adaptation of stream-dwelling fish. In oxygen-rich waters, these fish must not only efficiently utilize oxygen to enhance their metabolic capacity but also maintain redox homeostasis to prevent oxidative damage caused by high oxygen concentrations [5]. Additionally, high flow velocity requires the ability to swim continuously to resist the impacts of water flow and to sustain long-term exercise endurance [6]. Biological evidence has highlighted the morphological and physiological adaptations of fish to such specialized environments, including streamlined body shapes, well-developed red muscle tissues, and specialized fin structures [7,8]. However, the molecular mechanisms underlying the environmental adaptation of stream-dwelling fish remains limited. This gap primarily includes a lack of key genes involved in phenotypic adaptation, an incomplete exploration of genomic characteristics under environmental selection pressures, and insufficient information on the molecular pathways that facilitate adaptation to high-flow velocity and high-oxygen environments. These limitations have prevented a comprehensive and in-depth understanding of the environmental adaptation mechanisms in stream fishes.

Current research on *O. bidens* primarily focuses on its biology and ecology. Studies have confirmed that *O. bidens* exhibits significant adaptive responses to environmental changes, particularly in terms of body characteristics, life history strategies, and nutritional ecology [9,10,11]. At the molecular systematics level, existing studies primarily rely on mitochondrial sequences and limited nuclear gene markers. These studies have provided initial insights into the phylogenetic relationships [12], genetic diversity characteristics [13], and population historical dynamics [14]. However, due to the use of short fragment sequences, these studies face limitations in fully revealing the genomic structural characteristics and their associated environmental adaptation mechanisms. Although genomic studies have achieved chromosome-level assembly [15,16] and have begun to explore the genetic basis of sexual dimorphism [17], the relationship between karyotype polymorphism and Robertsonian translocation [18], there remains a significant gap in research of the molecular mechanisms underlying environmental adaptability of *O. bidens*. Notably, the existing genome assemblies still contain scaffold gaps and have not yet reached the complete T2T standard, particularly lacking precise annotations of key genomic structures such as telomeres and centromeres. Therefore, constructing a complete T2T-level reference genome will provide a crucial foundation for advancing adaptive evolution research at the whole-genome level.

In summary, *O. bidens*, as a representative stream-dwelling species, holds significant reference value for studies on environmental adaptability. However, the genetic basis of how this species adapts to stream environments remains unclear. To address this, the present study integrates PacBio HiFi, Nanopore ultra-long reads, and Hi-C technologies to comprehensively analyze the T2T-level genomic structural features of *O. bidens*, especially the complete information of telomeres and centromeres. Based on a comprehensive comparative genomics analysis, this study aims to systematically elucidate the genetic adaptations of *O. bidens* to high-dissolved-oxygen and high-flow-velocity stream environments. The results will provide essential resources and a theoretical basis for research on the adaptive evolution of stream fish and ecological genomics.

## 2. Materials and Methods

### 2.1. Sample Collection

In May 2023, a male specimen of *O. bidens* (126.42 mm in length, 32.5 g in weight, Figure 1B) was collected from the Jingde segment of the Huishui River in Anhui Province, China (30.274341° N, 118.530638° E, Figure 1A). The species in good condition was collected as part of the Monitoring of Aquatic Resources in Key Waters of Anhui Province Project (ZF2022-18-0399), under a special fishing permit ([2023]002) issued by the Department of Agriculture and Rural Affairs of Anhui Province. The specimen in good condition was randomly selected and anesthetized with MS-222. Under aseptic conditions, tissue samples-including muscle, liver, eyes, brain, spleen, gonads, gills, heart, and kidney-were dissected and immediately snap-frozen in liquid nitrogen. In subsequent procedures, DNA extraction was specifically performed on muscle and liver tissues to facilitate whole-genome assembly, while RNA was extracted from nine tissue samples for genome annotation.

### 2.2. DNA Extraction, Library Construction and Sequencing

Genomic DNA was first extracted for all samples. For MGI short-read and PacBio sequencing, genomic DNA was extracted using the CTAB method after grinding the tissue in liquid nitrogen. DNA quality was assessed by 0.75% agarose gel electrophoresis, NanoDrop spectrophotometry (Thermo Fisher Scientific, Waltham, MA, USA), and Qubit fluorometry (Thermo Fisher Scientific, Waltham, MA, USA). For Oxford Nanopore sequencing (ONT), high-molecular-weight DNA was extracted using the SDS method. DNA concentration and purity were measured by NanoDrop and Qubit, while DNA integrity was evaluated by pulsed-field gel electrophoresis. For Hi-C library preparation, tissue samples were crosslinked with 2% formaldehyde to preserve native chromatin conformation.

Library construction and sequencing were then performed. MGI short-read libraries were constructed using the MGIEasy Universal DNA Library Prep Kit V1.0 (CAT#1000005250, MGI) (MGI Tech Co. Ltd., Shenzhen, China) and sequenced on the DNBSEQ-T7 platform with 2 × 150 bp reads. Raw reads were filtered using fastp (v0.23.2) [19] with default parameters to remove low-quality reads, short reads, adapter sequences, and duplicates, retaining only paired-end reads longer than 50 bp. PacBio High Fidelity (HiFi) libraries were constructed with the SMRTbell^®^ prep kit 3.0 (PacBio) (Pacific Biosciences of California, Inc., Menlo Park, CA, USA) and sequenced on the Revio platform in Circular Consensus Sequencing (CCS) mode. CCS reads were generated using CCS v6.0.0 with default parameters to yield high-accuracy long reads. ONT ultra-long libraries were prepared using the SQK-LSK110 kit (Oxford Nanopore Technologies) (Oxford Nanopore Technologies plc, Oxford, UK) and sequenced on the Oxford Nanopore PromethION platform. High-throughput chromosome conformation capture (Hi-C) libraries were prepared using a combination of Biotin-14-dCTP (Invitrogen) (Thermo Fisher Scientific, Waltham, MA, USA), T4 DNA polymerase (NEB), and streptavidin magnetic beads (Invitrogen), and sequenced on the DNBSEQ-T7 platform with 2 × 150 bp reads.

### 2.3. Genome Assembly and Evaluation

Subsequently, genome assembly was carried out. Initial contigs were assembled from HiFi reads using Hifiasm (v0.19.6) (https://github.com/chhylp123/hifiasm, accessed on 23 July 2024) with the default purge_haplotigs module [20]. A non-redundant contig interaction matrix was generated using the HiCUP pipeline (v0.7.2) [21], followed by chromosome anchoring with the 3D-DNA workflow (180,922) [22]. Manual corrections of misassemblies such as inversions and translocations were performed using Juicebox Assembly Tools (1.11.08) [23]. TGS-GapCloser (v1.2.0) [24] was employed to fill the gaps between contigs by leveraging the coverage relationship between Oxford Nanopore ultra-long reads and pre-assembled contigs, thereby extending the contigs. Pilon (1.23) [25] was utilized to correct the extended and gap-filled genome using short-read sequencing data, resulting in a high-quality T2T-level genome assembly. The depths were calculated as the total data output divided by the genome size. Centromere regions were identified based on the distribution of tandem repeats using the multi-model prediction module in QuarTeT (1.2.1) [26].

Finally, the quality of the *O. bidens* genome assembly was evaluated in terms of accuracy, consistency, and completeness. To verify the correspondence between the assembly results and the target species, the genome was segmented into 10 kb fragments and aligned against the NCBI Nucleotide Database (NT database). The short-read data, PacBio long-read data, and ONT ultra-long-read data were mapped to the genome using bwa (0.7.12-r1039) [27] and minimap2 (2.24-r1122) [28] software, with sequence consistency evaluated based on alignment rate and coverage. The k-mer based consistency quality value was assessed using mercury (v1.3) [29]. The completeness of the genome assembly was evaluated using BUSCO (v5.7.1) [30] and Compleasm (v0.2.6) [31]. Both of them are based on the highly conserved OrthoDB database, which was constructed by sampling hundreds of genomes and selecting single-copy orthologous genes with a conservation rate exceeding 90% across six major phylogenetic branches.

### 2.4. Gene Prediction and Annotation

The annotation of repetitive sequences was conducted using a combined approach of homology-based prediction and de novo prediction. The former involved using RepeatMasker (v4.0.9) [32] and RepeatProteinMask (v4.1.0) (http://www.repeatmasker.org, accessed on 10 August 2024) to query the Repbase database [33] with the genomic sequence, followed by integration with the de novo transposable element (TE) library. The latter utilized LTR_FINDER_parallel (v1.0.7) [34] and RepeatModeler (v1.0.11) [35] to construct the repetitive sequence library.

Protein-coding gene annotation was performed using a comprehensive approach integrating ab initio prediction, homology comparison, and RNA-Seq-assisted annotation. Ab initio predictions were performed using Augustus (v3.3.2) [36] and Genscan [37], with species-specific models established based on the organism’s taxonomic classification. Homology-based predictions were executed using protein sequences from closely related species: *Danio rerio* (GenBank Assembly: GCF_000002035.6), *Hypophthalmichthys molitrix* (GenBank Assembly: 12618884), *Opsariichthys bidens* (GenBank Assembly: GWHBJYU00000000), and *Zacco platypus* (GenBank Assembly: GCA_034642465.1). Following TBlastN filtering (E-value ≤ 1 × 10^−5^), gene structures were refined using miniprot (v0.11-r234) [38] and liftoff (v1.6) [39]. RNA-Seq-assisted annotation was performed through transcriptome data integration from various tissues, with sequence alignment conducted using HISAT2 (v2.1.0) [40] and assembly performed with StringTie (v1.3.5) [41]. Finally, all gene models were consolidated and redundant entries were eliminated using MAKER2 (v2.31.10) [42] with default parameters and HiFAP (https://www.onemore-tech.com/, accessed on 30 August 2024).

Functional annotation of proteins was carried out based on sequence similarity and domain architecture. Gene sequences were aligned against multiple databases including NCBI non-redundant protein database (NR) [43], Swiss-Prot (http://www.gpmaw.com/html/swiss-prot.html, accessed on 17 September 2024), TrEMBL (http://www.uniprot.org, accessed on 17 September 2024), Eukaryotic Orthologous Groups (KOG) (https://ftp.ncbi.nih.gov/pub/COG/KOG/, accessed on 17 September 2024), and Kyoto Encyclopedia of Genes and Genomes (KEGG) (http://www.genome.jp/kegg/, accessed on 17 September 2024) using DIAMOND (v2.0.14) [44], with an E-value threshold of 1 × 10^−5^. Protein domain analysis was performed using InterProScan (v5.61-93.0) [45] and hmmscan3 (3.3.1) [46].

The identification of rRNA, miRNA, and snRNA were performed using INFERNAL based on Rfam (v14.8) [47] and miRBase (http://www.mirbase.org/, accessed on 6 November 2024) [48], while tRNA was annotated using tRNAscan-SE (v1.3.1) [49].

### 2.5. Genome Comparison and Identification of Newly Assembled Genes

The self-assembled genome of *O. bidens* was subjected to synteny analysis against the published genome (GCA_037194315.1) with a consistent karyotype (2n = 76) using the Syri (v0.1) [50] software. Chromosomal alignment between the two genome versions was performed using Winnomap (2.03) [51], followed by comprehensive comparison. BEDTools (v2.30.0) [51] was employed to identify and summarize previously unresolved regions (PURs) in the published genome. Corresponding genes within the PURs were extracted and subsequently analyzed for functional enrichment.

### 2.6. Gene Family and Phylogenomic Analysis

In addition to the self-assembled genome of *O. bidens*, the genome assembly and annotation data of 11 teleost species were retrieved from NCBI and Ensembl (Appendix A). For genes with multiple transcripts, the longest transcript was retained, while those encoding proteins shorter than 30 amino acids, containing internal stop codons, or exhibiting length errors were filtered out. OrthoMCL (v2.0.9) [52] (inflation parameter 1.5) was employed to cluster the results based on protein sequence similarity (E-value < 1 × 10^−5^). MAFFT (v7.525) [53] was subsequently utilized to align single-copy orthologous genes shared among all fish species. A maximum likelihood phylogenetic tree was then constructed from this alignment using RAxML (v8.2.12) [54] under the GTRGAMMA model with 1000 bootstrap replicates, rooted with *Gasterosteus aculeatus* (Appendix A). Based on the phylogenetic tree and divergence times, gene family expansion and contraction were analyzed in CAFE4 (v4.2) and CAFE5 (v5.0.0) [55,56] with a significance threshold of 0.05, followed by enrichment analysis. Divergence times were estimated using the MCMCMCTREE (v4.9) [55] program in PAML [57].

### 2.7. Gene Positive Selection Analysis

Multiple sequence alignment of gene-protein sequences within single-copy gene families was performed using the MAFFT software (v7.525) [53]. Likelihood ratio tests (LRTs) were conducted via the CodeML (v4.9) [55] module in PAML (*FDR* < 0.05). Subsequently, Bayes empirical Bayes method (BEB) was employed to detect positive selection acting on protein-coding sequences. Generally, sites with a posterior probability greater than 0.95 are considered to be significantly under positive selection.

## 3. Results and Discussion

### 3.1. Genome Sequencing and Gap-Free Assembly

K-mer analysis estimated the genome size of *O. bidens* to be approximately 847 Mb, with a heterozygosity rate of 0.49% (Figure 2 and Appendix A). By integrating PacBio HiFi, Oxford Nanopore ultra-long reads, and Hi-C technologies, we generated an initial assembly of 872.22 Mb, comprising a maximum contig length of 42.22 Mb and a contig N50 of 22.42 Mb (Appendix A). The depths of 32.97× for PacBio HiFi, 102.52× for ONT, and 100.72× for Hi-C sequencing. After scaffolding and anchoring, the final assembly was assigned to 38 chromosomes, with an anchoring rate of 98.09% (Appendix A). Hi-C interaction mapping (Figure 3B) revealed excellent intra-chromosomal signals along the diagonal and minimal inter-chromosomal noise in off-diagonal regions, supporting the accuracy of chromosome-level scaffold ordering and the absence of major misassemblies. Quality assessments using both short-read (99.94% genome coverage) and long-read data (99.96%) indicated high base-level accuracy (Appendix A). The homozygous rate of SNP and InDel was 0.001%, reflecting a highly accurate and contiguous genome assembly (Appendix A). Collectively, these results demonstrate that the assembled genome possesses strong structural integrity and high assembly quality (Figure 3A).

We generated a chromosome-scale, gap-free genome assembly of *O. bidens* (841.96 Mb) by using Oxford Nanopore ultra-long reads to bridge inter-contig gaps and extend contig lengths based on coverage relationships with pre-assembled contigs (Figure 4 and Table 1). Both the contig N50 and scaffold N50 reached 25.74 Mb. The identical values suggests that the assembled genome is gap-free, with no additional joins introduced during scaffolding. Taking advantage of the long-read capabilities of PacBio sequencing, we were able to assemble repetitive regions including centromeric and telomeric sequences. Telomeric repeats were detected at both ends of all chromosomes except for chr31, which may lack one telomere due to its single-ended configuration or highly repetitive content (Table 2). Centromere positions were identified using quarTeT-based predictions, Hi-C contact signal patterns, and element density profiles (Appendix A). The enhanced assembly continuity and completeness significantly improved our ability to resolve regions with high mutation rates. For example, in Gasterosteus aculeatus, PacBio sequencing increased genome continuity by more than fivefold, raising the N50 from 91.7 kb to 510.8 kb and filling gaps in highly repetitive regions such as telomeres and centromeres [58]. Furthermore, systematic evaluation of PacBio HiFi and ONT platforms using high-quality human reference specimens (HG002 and HG00733) demonstrated that structural variant (SV) detection based on high-quality assemblies substantially outperformed read-mapping-based methods in both sensitivity and precision, especially for SVs exceeding 10 kb at sequencing depths greater than 12× [59]. These findings highlight that improvements in genome assembly quality can markedly enhance the resolution of single nucleotide polymorphisms (SNPs) and SVs, particularly in structurally complex genomic regions.

Compared with the previously published genome assembly of *O. bidens* (GCA_037194315.1) [18], which shares the same karyotype (Figure 5), the newly assembled genome exhibits a high degree of synteny. Most chromosomal segments show clear collinearity, indicating strong syntenic conservation and substantial genomic integrity between the two assemblies. Furthermore, the gap-filled regions were significantly enriched in key pathways and functions. These include processes related to transcriptional regulation, protein degradation, mitochondrial metabolism, lipid transport, and immune responses (Appendix A). This enrichment indicates that these previously unassembled regions play critical roles in fundamental biological processes. Relative to the earlier version (Table 3), the contig N50 has increased from 9.01 Mb to 22.42 Mb, and the total number of contigs has been reduced from 98 to 38. Single-contig assembly was achieved for each chromosome. Meanwhile, the BUSCO completeness assessment reached 99.34% (Appendix A), which represents the current highest level, indicating that almost all functional gene regions were completely covered. Together, these improvements represent a substantial enhancement in genome continuity and completeness. Importantly, this study presents the first complete genome assembly for a representative species of the Opsariichthyinae subfamily, a phylogenetically controversial and ecologically significant lineage. The new assembly resolves previously ambiguous telomeric and centromeric regions. As a member of a basal lineage within Cyprinidae, this high-quality reference genome provides a valuable resource for studying the molecular basis of environmental adaptation and reconstructing the ancestral genome architecture of the family.

### 3.2. Genome Annotation and Repetitive Element Analysis

Repetitive element annotation revealed that 48.39% of the *O. bidens* genome consists of non-redundant transposable elements (TEs) (Appendix A). Among these, DNA transposons, long interspersed nuclear elements (LINEs), short interspersed nuclear elements (SINEs), and long terminal repeats (LTRs) accounted for 26.90%, 4.23%, 0.98%, and 11.02% of the genome, respectively. The proportion of repetitive sequences varies widely across teleost genomes. In evolutionarily conserved species such as *Siniperca kneri* (5.52%) and *S. chuatsi* (16.33%) [60], repetitive content tends to be low. In contrast, most teleost species maintain moderate levels of repetitive sequences, such as *Gadus morhua* (25.4%) [61] and *Gasterosteus aculeatus* (25.2%) [62]. In ancient polyploid taxa, repetitive elements may constitute a large proportion of the genome, as in Salmo salar (60%) [63]. The proportion of repetitive sequences in *O. bidens* genome thus represents a relatively high level among diploid teleosts and presumably may be related to genomic structural stability and sequence diversity.

Gene prediction based on both de novo and homology-based approaches identified 29,492 and 36,737 protein-coding genes, respectively. Homology-based predictions across four representative teleost genomes yielded gene counts ranging from 34,153 to 65,986. After integration and removal of redundant entries, a final non-redundant consensus gene set comprising 29,816 protein-coding genes was obtained, with an average gene length of 13,047 bp. Structural features of the annotated genes showed high consistency with those of closely related species. Functional annotations were successfully assigned to 27,169 genes (91.12%) using multiple public databases, providing a comprehensive foundation for downstream analyses of gene function, molecular pathways, and comparative genomic analyses (Appendix A).

Annotation of non-coding RNAs identified four major classes in the *O. bidens* genome, including 1165 microRNAs (miRNAs), 8681 transfer RNAs (tRNAs), 22,396 ribosomal RNAs (rRNAs), and 1698 small nuclear RNAs (snRNAs) (Appendix A). These annotations provide an essential foundation for understanding the regulatory landscape of gene expression in this species.

Gene set completeness was evaluated using both BUSCO and Compleasm, which identified 3542 (97.31%) and 3545 (97.39%) complete BUSCOs, respectively (Appendix A). These results indicate that the annotated gene set is highly complete and suiTable for downstream functional and evolutionary analyses.

### 3.3. Comparative Genomics Analysis

Based on gene family clustering (Figure 6A,B and Appendix A), 69 unique gene families, 6433 common gene families, and 2933 single-copy orthologs were identified. Using these single-copy genes, the representative support rate of each clade reached 100%, highlighting the effectiveness of single-copy genes in accurately resolving phylogenetic relationships. Using known species divergence times *G. aculeatus*–*C. molitorella* (252.2-180.8 MYA), *A. fasciatus*–*X. davidi* (124.7-81.0), *P. elongata*–*H. molitrix* (58.6-19.3 MYA), and *C. auratus* subA–*C. auratus* subB (21.5-17.5 MYA), were used as a correction time to construct phylogenetic trees and determine species developmental relationships (Appendix A). The results indicated that *O. bidens* and its closely related species *Z. platypus* shared the closest relationship, with an estimated divergence time of 13.5 MYA based on fossil records. Collinearity analysis further confirmed high homology between the two species (Figure 3C).

For such widely distributed species with strong ecological plasticity, adaptive evolution is often accompanied by complex changes in gene families and selection pressures [64]. Based on the above 12 species, gene family evolution was analyzed using CAFE (v5.0.0), revealing that 350 gene families underwent expansion and 584 gene families underwent contraction (Figure 6C, Appendix A). Positive selection analysis using CodeML (v4.9) [55] identified 193 positively selected genes (*FDR* < 0.05, Appendix A).

The expansion of gene families has provided a fundamental genetic reservoir for adaptation to high dissolved oxygen and high flow velocity in *O. bidens*. Based on annotation results, we observed significant gene enrichment in KEGG pathways associated with high dissolved oxygen adaptation, including the Phagosome pathway (ko04145, *p =* 1.3 × 10^−5^) and Oxidative phosphorylation (ko00190, *p =* 0.0747). Similarly, significantly expanded GO terms were identified, encompassing the Myosin complex (GO:0016459, *p* = 4.3 × 10^−4^), Mitochondrial respiratory chain complex IV (GO:0005751, *p =* 0.152), and Mitochondrial electron transport, cytochrome c to oxygen (GO:0006123, *p =* 0.469). Notably, the adaptation to high dissolved oxygen also emphasizes the repair of hyperoxic damage, requiring the clearance of damaged cellular components and regulation of redox balance to mitigate ROS accumulation. This phenomenon has been similarly reported in certain fish species inhabiting high-oxygen environments [65], suggesting a conserved adaptive strategy. These enrichment results reveal that *O. bidens* adapts to hyperoxic stress through antioxidant homeostasis, optimization of energy metabolism, and enhancement of mitochondrial function in high dissolved oxygen environments.

Furthermore, significant gene expansion pathways associated with flow velocity adaptation were identified, highlighting the need for efficient oxygen utilization to enhance metabolic capacity and the maintenance of redox homeostasis to mitigate hyperoxia-induced damage. The KEGG pathways revealed notable expansions in Cardiac muscle contraction (ko04260, *p* = 0.7278), Vascular smooth muscle contraction (ko04270, *p* = 0.9972), and Regulation of actin cytoskeleton (ko04810, *p* = 0.9972). Similarly, significantly enriched GO terms included Myosin complex (GO:0016459, *p* = 4.3 × 10^−4^) and Homophilic cell adhesion (GO:0007156, *p* = 6.3 × 10^−7^). These enrichment results demonstrate that *O. bidens* enhances its swimming performance and endurance in rapid-flow environments through strengthened muscle contraction, sustained swimming capacity, regulated cardiovascular energy supply, and dynamic cytoskeletal remodeling. While these mechanisms show certain similarities with those reported in migratory and other stream-dwelling fish species [65,66], the present study provides more comprehensive genomic-level evidence.

Concurrently, positive selection analysis conducted using the CodeML (v4.9) [55] software identified 193 positively selected genes (*FDR* < 0.05), revealing adaptive signals of *O. bidens* in high dissolved oxygen and high metabolic demand environments, primarily involving mitochondrial function optimization, oxygen utilization, and metabolic regulation [67,68,69]. These include pathways such as Propanoate metabolism (ko00640, *p* = 0.8146), Folate biosynthesis (ko00790, *p* = 0.8146), HIF-1 signaling pathway (ko04066, *p* = 0.8146), Prolactin signaling pathway (ko04917, *p* = 0.8146). In addition, genes such as *myo18a*, *slc25a51*, *slc25a25*, *mtx2*, *eral1*, *coq8b*, *th*, *mrpl21* further support its adaptive evolution in high-dissolved-oxygen and high-flow-velocity environments. Further functional validation is essential to confirm the adaptive roles of these genes and pathways.

Based on the T2T-level high-quality genome, we systematically analyzed the genetic adaptation characteristics of *O. bidens* in response to high-dissolved-oxygen and high-flow-velocity environments. The results showed that *O. bidens* has developed multidimensional adaptations during evolution, characterized by improved oxygen utilization efficiency and optimized continuous locomotion capacity, consistent with the features of stream environments. It should be noted that although this study identified partial gene family expansions and positive selection signals related to metabolic regulation and the motor system in *O. bidens*, whether these genomic changes are directly associated with adaptation to high dissolved oxygen and high-flow conditions requires further confirmation through functional validation, comparative population data, and environmental adaptation experiments. These findings provide preliminary clues for understanding the genetic adaptation characteristics of stream-dwelling fish and contribute to further exploration of the ecological adaptation mechanisms of fish in complex hydrological environments.

## 4. Conclusions

In this study, we constructed and updated the complete genome of *O. bidens*, and further discussed the genomic characteristics of *O. bidens* as a typical representative of stream environmental adaptation. Through the dual evolutionary mechanism of gene family expansion and positive selection, *O. bidens* has gained significant adaptive advantages in oxygen metabolism efficiency and locomotor capacity, enabling successful adaptation to high-dissolved-oxygen and high-flow-velocity stream environments. Our results provide a theoretical basis for the assessment of *O. bidens* germplasm resources and for studying the environmental adaptability of stream fish under complex ecological pressures, offering genomic resource for understanding biodiversity maintenance and ecosystem stability.

## Figures and Tables

**Figure 1 biology-14-01544-f001:**
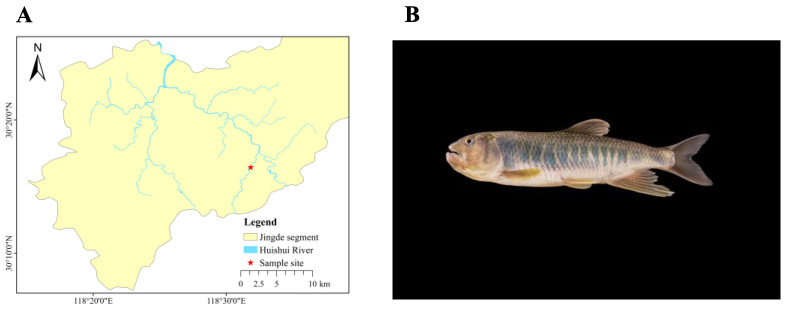
Specimen Data. (**A**). Map of sampling sites. (**B**). Live photo of *O. bidens*.

**Figure 2 biology-14-01544-f002:**
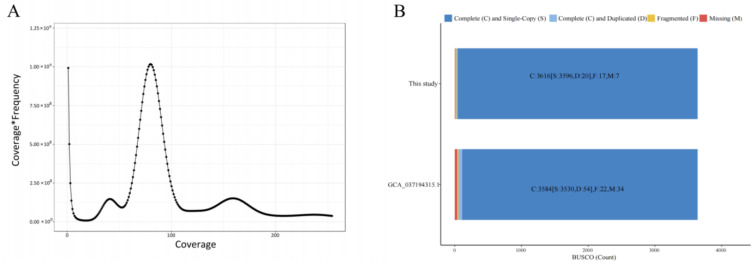
Assessment of genome assembly quality. (**A**) K-mer distribution frequency. (**B**) BUSCO assessment.

**Figure 3 biology-14-01544-f003:**
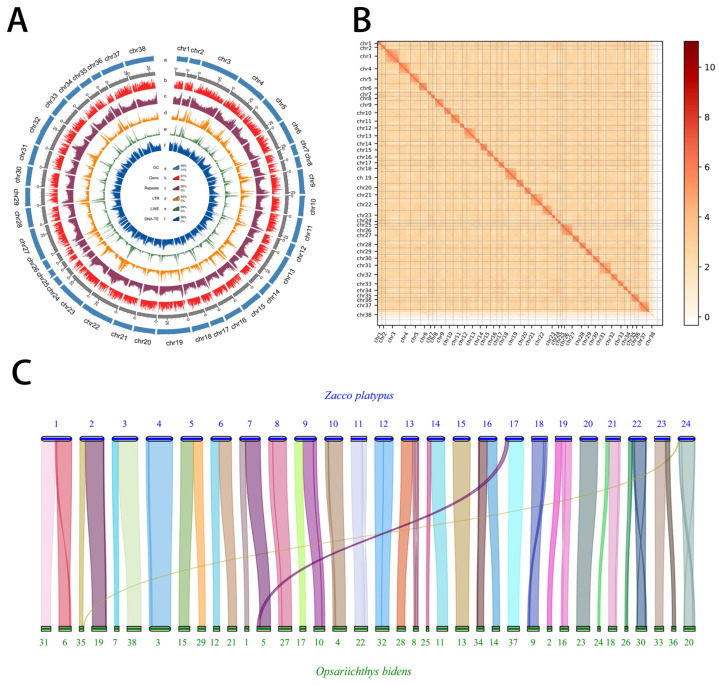
Genome assembly and gene annotation of *O. bidens*. (**A**) Features the *O. bidens* genome arranged from the outermost. (**B**) A heatmap of chromosomal interactions in *O. bidens*. (**C**) Synteny between genomes of *O. bidens* and *Zacco platypus*.

**Figure 4 biology-14-01544-f004:**
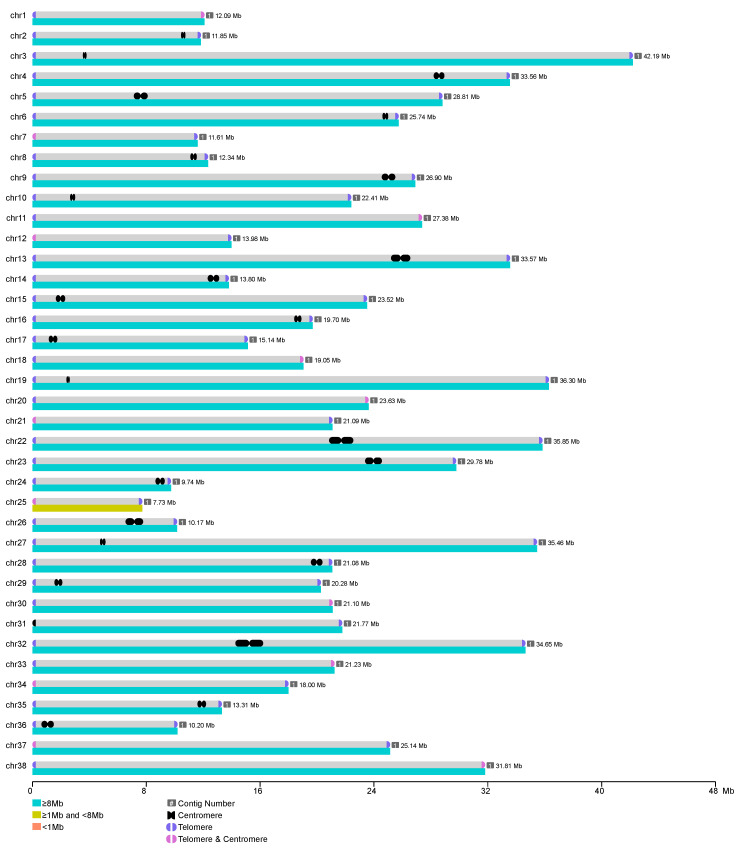
Visualization of T2T genome. Purple in the figure represents telomeres, black squares represent centromeres, and pink represents subtelomeric centromeres.

**Figure 5 biology-14-01544-f005:**
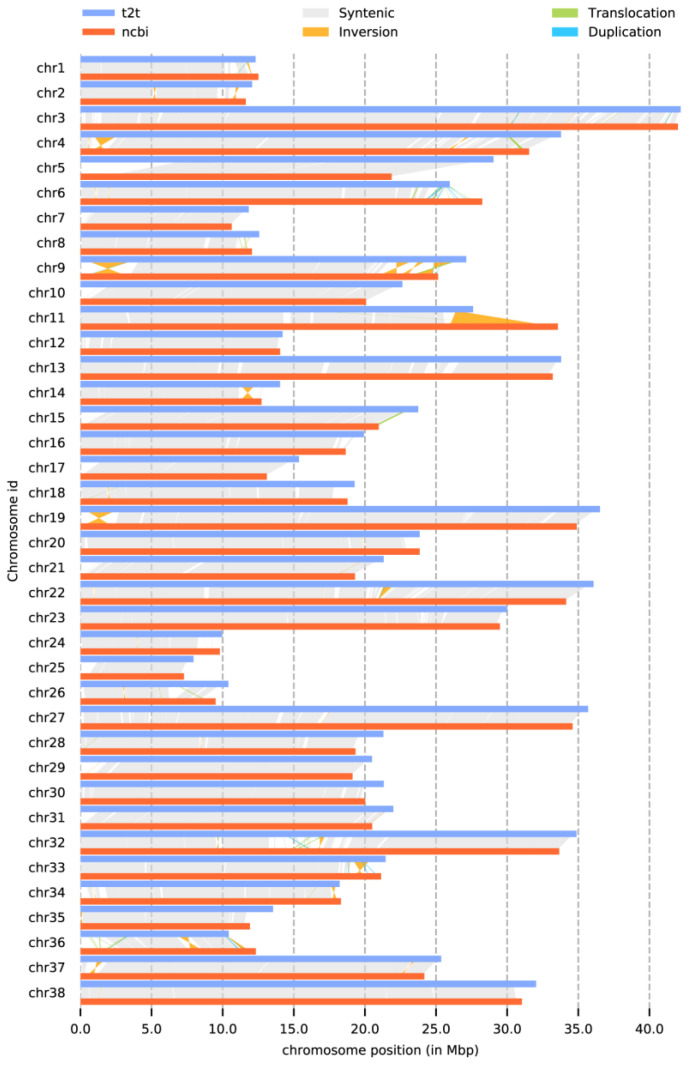
Collinearity analysis of different genome versions.

**Figure 6 biology-14-01544-f006:**
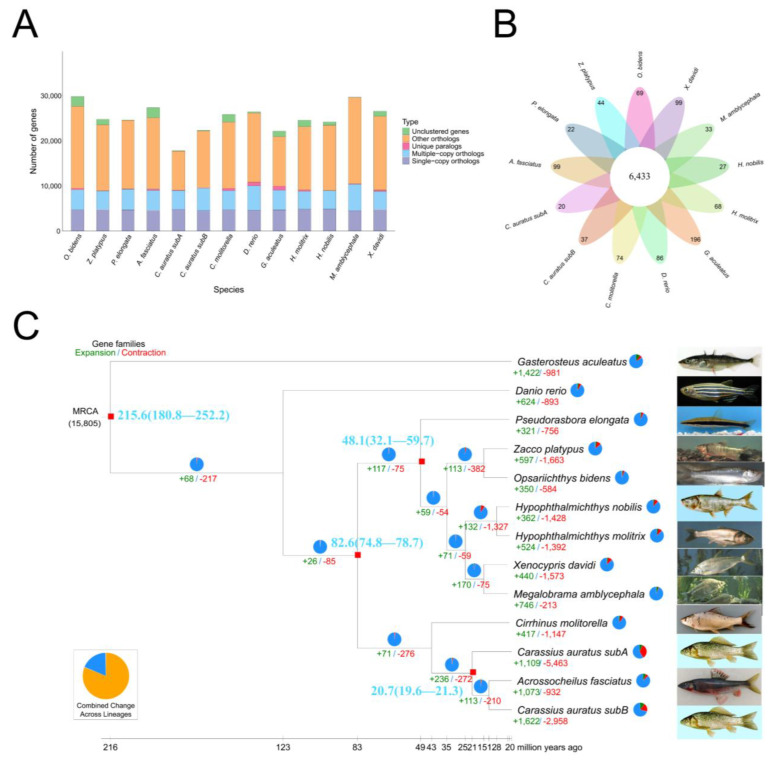
Comparative genomic analysis for *O. bidens*. (**A**) Orthologous gene families. (**B**) Unique and shared gene families. (**C**) Phylogenetic tree. The green numbers indicate expanded gene families, while the red numbers indicate contracted gene families.

**Table 1 biology-14-01544-t001:** Summary statistics of *O. bidens* genome assembly.

	Contig Length (bp)	Contig Number	Scaffold Length (bp)	Scaffold Number
N90	12,343,832	31	12,343,832	31
N80	19,053,543	25	19,053,543	25
N70	21,089,684	21	21,089,684	21
N60	22,405,681	17	22,405,681	17
N50	25,739,500	13	25,739,500	13
Total length	841,960,764	NA	841,960,764	NA
Number (≥100bp)	NA	38	NA	38
Number (≥2kb)	NA	38	NA	38
Max length	42,194,854	NA	42,194,854	NA

NA: Not Applicable (indicating that a field or parameter is irrelevant in the given context).

**Table 2 biology-14-01544-t002:** The details of chromosome and number of telomere characteristic sequences for *O. bidens* genome.

Chromosome	Length (bp)	Number of Contigs	Number of Gaps	Number of Start Repeat Unit	Number of End Repeat Unit
chr1	12,090,187	1	0	1878	2521
chr2	11,846,864	1	0	2173	478
chr3	42,194,854	1	0	2308	1440
chr4	33,560,156	1	0	2598	2236
chr5	28,811,627	1	0	725	1437
chr6	25,739,500	1	0	6251	23
chr7	11,611,510	1	0	4712	1556
chr8	12,343,832	1	0	1890	2057
chr9	26,899,604	1	0	1771	1723
chr10	22,405,681	1	0	1898	1442
chr11	27,380,574	1	0	1375	2676
chr12	13,984,826	1	0	1318	1118
chr13	33,569,505	1	0	1953	1989
chr14	13,802,948	1	0	1689	2244
chr15	23,521,344	1	0	4962	1668
chr16	19,700,785	1	0	3074	32
chr17	15,140,455	1	0	1221	674
chr18	19,053,543	1	0	1940	1682
chr19	36,299,173	1	0	3630	1463
chr20	23,626,312	1	0	2530	41
chr21	21,089,684	1	0	33	1642
chr22	35,847,689	1	0	1872	2997
chr23	29,779,289	1	0	2608	3403
chr24	9,738,410	1	0	1417	1527
chr25	7,725,830	1	0	2161	1856
chr26	10,171,860	1	0	963	1397
chr27	35,461,429	1	0	2234	1577
chr28	21,077,629	1	0	1723	1847
chr29	20,275,876	1	0	45	2229
chr30	21,102,698	1	0	3138	7
chr31	21,769,779	1	0	0	1951
chr32	34,646,360	1	0	1195	2618
chr33	21,231,445	1	0	3229	1936
chr34	18,000,321	1	0	9	3372
chr35	13,313,169	1	0	1688	3629
chr36	10,202,237	1	0	1736	1113
chr37	25,135,100	1	0	78	1344
chr38	31,808,679	1	0	1458	2

**Table 3 biology-14-01544-t003:** Comparison of genomes of different versions of *O. bidens*.

Statistical Indicators	GCA_037194365.1	GCA_037194315.1	GCA_037194245.1	GWHBEIO00000000-1	GWHBEIO00000000-2	This Study
Sex	male	female	female	female	male	female
Total size of assembled genome (Mb)	852.41	843.11	840.94	818.78 Mb	992.9	841.96
Contig N50	9.01	2.9	5.27	4.66	5.2	25.74
Contig N90	NA	NA	NA	NA	NA	12.34
Number of Contigs	NA	NA	NA	403	1373	38
Scaffold N50 (Mb)	21.01	23.62	24.75	25.29	19.44	25.74
Scaffold N90 (Mb)	NA	NA	NA	NA	NA	12.34
Scaffold number	228	450	98	39	38	38
chromosome number	37	38	39	39	38	38
Number of gap-free chromosome	NA	NA	NA	NA	NA	38
Number of gaps	NA	NA	NA	NA	NA	0
Number of telomeres(pairs/single)	NA	NA	NA	NA	NA	37/1
TE size	360.05	356.29	356.64	347.06	357.31	407.46
GC content	38.3	38.3	38.1	NA	37.9	38.4
Chromosome anchoring ratio (%)	90.39	95.67	99.01	95.66%	89.31	98.09
Total chromosome length (Mb)	770.52	806.65	832.70	814.71	886.81	841.96
Gene number	26,556	25,036	26,283	23,992	36,738	29,816
Functional proteins	25,383 (95.58%)	23,139 (92.42%)	24,493 (93.19%)	22,869 (95.4%)	30,922 (84.17%)	27,169 (91.12%)
BUSCO completeness genome (%)	97.2	96.6	96.8	96.6	97.5	99.34
BUSCO completeness (protein)	91.7	91.8	94.5	93.5	NA	97.31
DNA matching rate (%)	98.78	99.05	98.98	98.76	99.5	99.94

NA: Not Available (indicating the absence of data or non-reporting).

## Data Availability

The sequencing dataset and genome assembly of *O. bidens* have been deposited in the NCBI SRA database under project number PRJNA1306202. The data are as follows: Hi-C data (SRR34997172); DNBSEQ-T7 genome sequencing data (SRR34997171); PacBio Revio genome sequencing data (SRR34997170); OXFORD_NANOPORE genome sequencing data (SRR34997168). The assembled genome was deposited in the NCBI Genome with the accession number GCA_046055825. Genome annotations, along with predicted coding sequences and protein sequences, can be accessed through the Figshare (https://doi.org/10.6084/m9.figshare.29848886 (accessed on 28 October 2025)).

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
