# Peer review of "Telomere-to-Telomere Gap-Free Genome Assembly and Comparative Analysis of the Opsariichthys bidens (Cypriniformes: Xenocyprididae)"

_biology, 2025, doi:10.3390/biology14111544_

Round 1

Reviewer 1 Report

Comments and Suggestions for Authors

This study presents a telomere-to-telomere (T2T), gap-free genome assembly for the stream-dwelling fish Opsaritchthys bidens. The integration of PacBio HiFi, Oxford Nanopore Ultra-long, and Hi-C technologies has resulted in a genome assembly of exceptional continuity and completeness, a significant improvement over several previous versions. The subsequent comparative genomic analysis provides insights into the genetic adaptations underlying this species' survival in high-dissolved-oxygen and high-flow-velocity environments. The manuscript is well-structured, the analyses are generally robust, and the work represents a valuable genomic resource for the field. I recommend acceptance after minor revisions.

Line 19, delete “first”,

Line 30, remove the second “a”

Line 63, remove “current knowledge of”

Line 83, delete a period.

Line 86, constructing a complete T2T-level reference genome is better.

Line 91, remain change to remains

Line 107, “The species in good condition……” should be rephrased.

Line 132, High-throughput, not high-through

Line 146, is it T2T level or chromosome-level?

Line 196, any citation for the Syri software?

Line 228, delete “successfully”

Line 235, delete “only”

Line 237, change “strong” into “excellent”

Line 238, rephrase the sentence to make it more straightforward, and be modest to use “successfully” in the whole manuscript.

Line 242, suggests,

Line 361-369, the sentence is too long to read, cut it into short sentences is suggested. Myo18a, slc25a51, where are the genes from? Why they could support the adaptive evolution in high dissolved oxygen? Sentence like “Further functional validation is essential” could be added.

The discussion would be stronger with a more direct comparison to other known stream-adapted or migratory fish (e.g., salmonids). Are the same pathways and gene families under selection, or does O. bidens show a unique suite of adaptations?

Author Response

Reviewer #1:

This study presents a telomere-to-telomere (T2T), gap-free genome assembly for the stream-dwelling fish Opsaritchthys bidens. The integration of PacBio HiFi, Oxford Nanopore Ultra-long, and Hi-C technologies has resulted in a genome assembly of exceptional continuity and completeness, a significant improvement over several previous versions. The subsequent comparative genomic analysis provides insights into the genetic adaptations underlying this species' survival in high-dissolved-oxygen and high-flow-velocity environments. The manuscript is well-structured, the analyses are generally robust, and the work represents a valuable genomic resource for the field. I recommend acceptance after minor revisions.

Response:Thank you so much for your positive feedback on our manuscript. We truly appreciate your kind words and support. We have meticulously addressed issues raised. All modifications have been thoroughly integrated into the revised manuscript. Our point-by-point responses to each comment are detailed below.

Comments 1: Line 19, delete “first”

We thank the reviewer for this suggestion and have updated the text to: In this study, we constructed the telomere-to-telomere (T2T) reference genome for O. bidens.

Comments 2: Line 30, remove the second “a”

We thank the reviewer for this suggestion and have updated the text to: The assembled genome spans 841.96 Mb, comprising 38 chromosomes, each in a single contig (contig N50 = 22.42 Mb, 2.5-fold higher than the previous version), achieving a gap-free standard with 99.34% BUSCO completeness.

Comments 3: Line 63, remove “current knowledge of”

We thank the reviewer for this suggestion and have updated the text to: However, the molecular mechanisms underlying the environmental adaptation of stream-dwelling fish remains limited.

Comments 4: Line 83, delete a period.

We thank the reviewer for this suggestion and have updated the text.

Comments 5: Line 86, constructing a complete T2T-level reference genome is better.

We thank the reviewer for this suggestion and have updated the text to: Therefore, constructing a complete T2T-level reference genome will provide a crucial foundation for advancing adaptive evolution research at the whole-genome level.

Comments 6: Line 91, remain change to remains

We thank the reviewer for this suggestion and have updated the text to: However, the genetic basis of how this species adapts to stream environments remains unclear.

Comments 7: Line 107, “The species in good condition……” should be rephrased.

We thank the reviewer for this suggestion and have updated the text to: The species in good condition was collected as part of the Monitoring of Aquatic Resources in Key Waters of Anhui Province Project.

Comments 8: Line 132, High-throughput, not high-through

We thank the reviewer for this suggestion and have updated the text to: High-throughput chromosome conformation capture (Hi-C) libraries were prepared...

Comments 9: Line 146, is it T2T level or chromosome-level?

We thank the reviewer for this pertinent question. We confirm that our assembly has indeed achieved a T2T-level standard. The original phrasing has been revised to remove any ambiguity and to consistently use the accurate terminology throughout the manuscript. The revised sentence now reads: "Pilon was utilized to correct the extended and gap-filled genome using short-read sequencing data, resulting in a high-quality T2T-level genome assembly."

Comments 10: Line 196, any citation for the Syri software?

We thank the reviewer for pointing this out. As suggested, we have now added the appropriate citation for Syri software.

Comments 11: Line 228, delete “successfully”

We thank the reviewer for this suggestion and have updated the text to: After scaffolding and anchoring, the final assembly was assigned to 38 chromosomes.

Comments 12: Line 235, delete “only”

We thank the reviewer for this suggestion and have updated the text to: The homozygous rate of SNP and InDel was 0.001%, reflecting a highly accurate and contiguous genome assembly.

Comments 13: Line 237, change “strong” into “excellent”

We thank the reviewer for this suggestion and have updated the text to: Hi-C interaction mapping (Figure 3B) revealed excellent intra-chromosomal signals along the diagonal and minimal inter-chromosomal noise in off-diagonal regions.

Comments 14: Line 238, rephrase the sentence to make it more straightforward, and be modest to use “successfully” in the whole manuscript.

We thank the reviewer for this suggestion and have updated the text to: We generated a chromosome-scale, gap-free genome assembly of O. bidens (841.96 Mb) by using Oxford Nanopore ultra-long reads to bridge inter-contig gaps and extend contig lengths based on coverage relationships with pre-assembled contigs.

Comments 15: Line 242, suggests,

We thank the reviewer for this suggestion and have updated the text to: The identical values suggests that the assembled genome is gap-free, with no additional joins introduced during scaffolding.

Comments 16: Line 361-369, the sentence is too long to read, cut it into short sentences is suggested.

We thank the reviewer for this suggestion and have updated the text to: Compared with the previously published genome assembly of O. bidens (GCA_037194315.1), which shares the same karyotype (Figure 5), the newly assembled genome exhibits a high degree of synteny. Most chromosomal segments show clear collinearity, indicating strong syntenic conservation and substantial genomic integrity between the two assemblies. Furthermore, the gap-filled regions were significantly enriched in key pathways and functions. These include processes related to transcriptional regulation, protein degradation, mitochondrial metabolism, lipid transport, and immune responses (Table S8). This enrichment indicates that these previously unassembled regions play critical roles in fundamental biological processes.

Comments 17: Myo18a, slc25a51, where are the genes from? Why they could support the adaptive evolution in high dissolved oxygen? Sentence like “Further functional validation is essential” could be added.

Thank you for the comment. The genes (e.g., myo18a, slc25a51) are from our positive selection analysis in O. bidens. We highlighted them based on their annotated functions relevant to mitochondrial efficiency and oxygen response, suggesting a potential adaptive role. As suggested, we have added the statement: "Further functional validation is essential to confirm the adaptive roles of these genes and pathways".

Comments 17: The discussion would be stronger with a more direct comparison to other known stream-adapted or migratory fish (e.g., salmonids). Are the same pathways and gene families under selection, or does O. bidens show a unique suite of adaptations?

We thank the reviewer for the valuable suggestion to compare our findings with migratory fish (e.g., salmonids). We fully agree that comparing different life-history strategies is a crucial perspective for understanding flow adaptation. The core of our current study is focused on the "hydrological gradient in plain waters". To purely dissect the genomic adaptations to water flow velocity, we specifically selected representative species from stagnant, moderate-flow, and rapid-flow habitats (three species each), including a model species and an outgroup, for a controlled comparison.

We believe that the proposed comparison with migratory fish represents a significant scientific question that is more suitably addressed in an independent, future study. We have therefore established this as a clear direction for our subsequent research. We appreciate the reviewer's insightful guidance.

Reviewer 2 Report

Comments and Suggestions for Authors

The authors have put in good efforts in data description and discussion on telomere-to-telomere gap-free genome assembly and comparative analysis of the Opsariichthys bidens (Cypriniformes: Xenocyprididae). The methodology presented is highly reproducible. It is good if authors can include bootstrapping in the phylogenetic tree presented.

The English language usage is professional and standard for journal publication. The methodologies utilized is of high standard and replicable, consistent with other researches in the same subject area. It is also good to provide the original tree with branch length varied to show more information in phylogenetic tree if possible (with 1000 bootstrap replications and Maximum likelihood or appropriate algorithms and models) and provide appropriate outgroups and scale bar.  The discussions are in depth and can be improved further by adding more up-to-date recent references and citations (from 2020 - 2025).

Thank you. 

Author Response

Reviewer #2:

The authors have put in good efforts in data description and discussion on telomere-to-telomere gap-free genome assembly and comparative analysis of the Opsariichthys bidens (Cypriniformes: Xenocyprididae). The methodology presented is highly reproducible. It is good if authors can include bootstrapping in the phylogenetic tree presented.

The English language usage is professional and standard for journal publication. The methodologies utilized is of high standard and replicable, consistent with other researches in the same subject area. It is also good to provide the original tree with branch length varied to show more information in phylogenetic tree if possible (with 1000 bootstrap replications and Maximum likelihood or appropriate algorithms and models) and provide appropriate outgroups and scale bar. The discussions are in depth and can be improved further by adding more up-to-date recent references and citations (from 2020 - 2025).

Response:We thank the reviewer for this suggestion. We have now replaced the original phylogenetic tree (Figure S5) in the supplementary materials with a comprehensively updated version. The new tree was reconstructed using Maximum Likelihood method in RAxML with 1,000 bootstrap replicates to assess branch support. It now displays branch lengths proportional to evolutionary distance, is rooted with an appropriate outgroup Gasterosteus aculeatus, and includes a scale bar. This updated figure provides full statistical and evolutionary information. The original calibration nodes previously shown in Figure S5 are already represented in Figure 6 of the main text, thus this replacement optimizes the presentation without loss of key information. We have updated the text to: MAFFT (v7.525) was subsequently utilized to align single-copy orthologous genes shared among all fish species. A maximum likelihood phylogenetic tree was then constructed from this alignment using RAxML (v8.2.12) under the GTRGAMMA model with 1,000 bootstrap replicates, rooted with Gasterosteus aculeatus (Figure S5). 

And we thank the reviewer for this insightful suggestion. Following the reviewer's advice, we have added two recent references (from 2023 and 2021) concerning dissolved oxygen adaptation to the discussion. These additions have significantly strengthened the depth and contemporary relevance of our discussion on this topic.

Reviewer 3 Report

Comments and Suggestions for Authors

This study conducted by Liu et al. (2025) introduces “Telomere-to-telomere gap-free genome assembly and comparative analysis of the Opsariichthys bidens (Cypriniformes: Xenocyprididae)”. The authors constructed the first telomere-to-telomere (T2T) reference genome for O. bidens, a freshwater fish uniquely adapted to challenging stream environments. They show how a high quality genome assembly provides a foundational resource for understanding the genetic mechanisms underlying ecological adaptation in stream-dwelling fishes.

Although the authors present a standard approach to build the T2T assembly, the study represents a valuable step toward environmental adaptability, biodiversity maintenance and ecosystem stability.

Comments:

  1. The overall assembly quality is currently presented in Tables 1, S2, and S3. The manuscript lacks a comprehensive main figure illustrating the principal assembly statistics. I recommend expanding Figure 2 to include analysis of assembly quality metrics following the scaffolding and anchoring steps:
  • Figure 2a: Retain the current Figure 2, but add axis x and y labels for clarity.
  • Figure 2b: Include an NGx plot displaying contig/scaffold lengths across different percentages of the total assembled genome length. This will provide a comprehensive view of assembly contiguity and completeness. Please refer to Figure 1D in Liao et al. for examples of such visualizations.
  • Figure 2c: Add a standard BUSCO assessment plot. 

If possible, include statistics for the previously published O. bidens genome assembly (GCA_037194315.1) in this expanded Figure 2.

  • Sequencing Depth Information. Page 3, "DNA extraction, Library construction and sequencing" section: Please specify the sequencing depth for ONT, PacBio HiFi, and Hi-C in both the main text and methods section.

  • Software Version Numbers. Page 4, line 198: Add the version number for Winnowmap. Please verify throughout the entire manuscript that version numbers are provided for all bioinformatics tools and software used.

  • GCA_037194315.1. Page 6, line 261. Please add the reference to the previous genome assembly.

  • Data Availability Statement. The "Data Availability" section must clearly specify: (1) data availability used in the study and access location (2) a functional Figshare link.

Author Response

Reviewer #3:

This study conducted by Liu et al. (2025) introduces “Telomere-to-telomere gap-free genome assembly and comparative analysis of the Opsariichthys bidens (Cypriniformes: Xenocyprididae)”. The authors constructed the first telomere-to-telomere (T2T) reference genome for O. bidens, a freshwater fish uniquely adapted to challenging stream environments. They show how a high quality genome assembly provides a foundational resource for understanding the genetic mechanisms underlying ecological adaptation in stream-dwelling fishes.

Although the authors present a standard approach to build the T2T assembly, the study represents a valuable step toward environmental adaptability, biodiversity maintenance and ecosystem stability.

Response:Thank you so much for your positive feedback on our manuscript. We truly appreciate your kind words and support. We have meticulously addressed issues raised. All modifications have been thoroughly integrated into the revised manuscript. Our point-by-point responses to each comment are detailed below.

Comments 1: The overall assembly quality is currently presented in Tables 1, S2, and S3. The manuscript lacks a comprehensive main figure illustrating the principal assembly statistics. I recommend expanding Figure 2 to include analysis of assembly quality metrics following the scaffolding and anchoring steps.

We thank the reviewer for the valuable suggestions to improve Figure 2. We have now expanded it into a comprehensive visual summary of assembly quality. Specifically, we have retained and clarified Figure 2a with axis labels and introduced a new Figure 2b featuring a comparative BUSCO analysis against the published O. bidens genome. These revisions provide a clear and standardized overview of our assembly's key metrics.

  • Figure 2a: Retain the current Figure 2, but add axis x and y labels for clarity.

Regarding Figure 2a: We thank the reviewer for the comment. The X and Y axis labels have been added to Figure 2a to improve the clarity of the figure, as recommended.

  • Figure 2b: Include an NGx plot displaying contig/scaffold lengths across different percentages of the total assembled genome length. This will provide a comprehensive view of assembly contiguity and completeness. Please refer to Figure 1D in Liao et al. for examples of such visualizations.

Regarding Figure 2b: We thank the reviewer for highlighting the value of the NGx plot for visually comparing assembly contiguity. We recognize it provides a holistic view that our other figures do not directly replicate. However, due to the unavailability of the requisite intermediate files, regenerating it is not feasible. We have therefore focused on strengthening the contiguity assessment with the available data. Table 3 now provides an expanded set of metrics, including not only our superior Contig N50 (25.74 Mb), but also the Contig N90 (12.34 Mb) and the dramatically lower number of contigs (38), which together offer unambiguous quantitative evidence of a more continuous and gapless assembly at the base level. This, combined with the chromosome-level completeness shown in Figure 5, we believe, compellingly demonstrates the advanced contiguity of our genome.

  • Figure 2c: Add a standard BUSCO assessment plot.If possible, include statistics for the previously published bidens genome assembly (GCA_037194315.1) in this expanded Figure 2.

Regarding Figure 2c: We thank the reviewer for this excellent suggestion. We have now performed a BUSCO analysis for both our new genome assembly and the previously published O. bidents assembly (GCA_037194315.1). The updated figure clearly demonstrates that our assembly achieves a higher level of completeness, with a significant increase in the proportion of complete BUSCO genes compared to the previously published genome. This provides strong, standardized evidence for the improved quality of our genomic resource. This plot has been incorporated as the new Figure 2b.

Comments 2: Sequencing Depth Information. Page 3, "DNA extraction, Library construction and sequencing" section: Please specify the sequencing depth for ONT, PacBio HiFi, and Hi-C in both the main text and methods section.

We thank the reviewer for this suggestion. In response, we have now specified the sequencing depths in both the main text and the Methods section. The depths were calculated as the total data output divided by the genome size, yielding values of 32.97× for PacBio HiFi, 102.52× for ONT, and 100.72× for Hi-C sequencing.

Comments 3: Software Version Numbers. Page 4, line 198: Add the version number for Winnowmap. Please verify throughout the entire manuscript that version numbers are provided for all bioinformatics tools and software used.

We thank the reviewer for this comment. We have now added the version number for Winnowmap, and have carefully verified and updated the version numbers for all bioinformatics tools and software throughout the entire manuscript.

Comments 4: GCA_037194315.1. Page 6, line 261. Please add the reference to the previous genome assembly.

We thank the reviewer for the suggestion. The reference for the previous genome assembly (GCA_037194315.1) has been added.

Comments 5: Data Availability Statement. The "Data Availability" section must clearly specify: (1) data availability used in the study and access location (2) a functional Figshare link.

We have updated the Data Availability Statement in accordance with the reviewer's suggestion. It now includes: The sequencing dataset and genome assembly of O. bidens have been deposited in the NCBI SRA database under project number PRJNA1306202. The data are as follows: Hi-C data (SRR34997172); DNBSEQ-T7 genome sequencing data (SRR34997171); PacBio Revio genome sequencing data (SRR34997170) ; OXFORD_NANOPORE genome sequencing data (SRR34997168). The assembled genome was deposited in the NCBI Genome with the accession number GCA_046055825. Genome annotations, along with predicted coding sequences and protein sequences, can be accessed through the Figshare (10.6084/m9.figshare.29848886).

Round 2

Reviewer 3 Report

Comments and Suggestions for Authors

The revised manuscript has been carefully evaluated, and I am satisfied that you have adequately addressed all the concerns raised by the  initial review process. The revisions you have implemented have significantly strengthened the manuscript, and both the methodology and presentation have been improved.